# Effects of Wearing FFP2 Masks on SARS-CoV-2 Infection Rates in Classrooms

**DOI:** 10.3390/ijerph192013511

**Published:** 2022-10-19

**Authors:** Gerald Jarnig, Reinhold Kerbl, Mireille N. M. van Poppel

**Affiliations:** 1Institute of Human Movement Science, Sport and Health, University of Graz, 8010 Graz, Austria; 2Department of Pediatrics and Adolescent Medicine, LKH Hochsteiermark, 8700 Leoben, Austria

**Keywords:** SARS-CoV-2, COVID-19, masks, school, children, transmission, infection, FFP2

## Abstract

In this retrospective cohort study involving 614 secondary school students, the likelihood of becoming infected with SARS-CoV-2 in schools with different focus (sports focus vs. general branch; the only difference in the sports focus school was that PE was allowed at all times without restrictions) and different prevailing restrictions were compared. A significantly higher likelihood of infection with SARS-CoV-2 was found in sports classes during the period with a strict FFP-2 mask requirement compared to general branch classes (for Delta from November 2021 to December 2021, and for Omicron from January 2022 to February 2022). The higher likelihood of infection was observed both during the Delta and the Omicron wave. After the relaxation of the mitigation measures, however, students in general branch classes showed a clear “catch-up” of infections, leading to a higher incidence of infections during this phase. By the end of the observation period (30 April 2022), only a small difference in cumulative SARS-CoV-2 infection rates (*p* = 0.037, φ = 0.09) was detected between classes with a sports focus and those without a sports focus. The results suggest that SARS-CoV-2 transmission can be reduced in school classes by mandatory FFP-2 mask use. In many cases, however, infection appears to be postponed rather than avoided.

## 1. Introduction

Worldwide, different restrictions were mandated to control the spread of different severe acute respiratory syndrome coronavirus 2 (SARS-CoV-2) variants. The stringency of these was based on the predominant viral load in the national population [1]. As one of these measures, the usefulness and practicality of medical face masks to contain the spread of viruses has been proven [2,3]. Increasing numbers of studies are reporting the usefulness of FFP2 masks to reduce the spread of SARS-CoV-2 [4,5,6]. In public spaces, in addition to strict hygiene rules and a minimum safety distance, the use of face masks has been recommended as they have been shown to greatly reduce the spread of different virus variants [7]. In June 2020, a recommendation was published by the World Health Organization (WHO) describing the use of face masks (mouth–nose masks) in the context of coronavirus disease 2019 (COVID-19) [8]. On 21 August 2020, this recommendation was expanded to detail describing how children should use a face mask in the community [9]. Thus far, no negative physical effects of face masks have been proven for children [10,11,12]. However, an increasing number of studies report psychological problems triggered by continued mandatory use of these [13,14,15]. Therefore, it appears essential to carefully balance the advance of reduced virus transmission against the negative behavioral and psychological aspects, as well as in the light of concerns about the correct wearing of face masks by children over a long period [16,17,18].

The aim of this study was to assess if wearing face masks in the classroom is a useful mitigation measure to control SARS-CoV-2 infections and to gain an impression whether this effect is big enough to override potential negative “side effects”.

## 2. Materials and Methods

### 2.1. Design

In this retrospective cohort study, we compared cumulative SARS-CoV-2 infection rates in sports classes (with limited use of face masks) and non-sports classes (with the consequent use of face masks). The study was registered in the German Clinical Trial Register (ID DRKS00029061) and approved by the Research Ethics Committee at the University of Graz, Styria, Austria (GZ. 39/70/63 ex 2021/22).

### 2.2. Selection of Schools and Participation

A school campus in Klagenfurt, Austria, was selected, consisting of a secondary school with a focus on “development for competitive sports”, and a parallel general school branch (GB). Children/adolescents attending the branch with a sports focus (SF) were allowed to participate in sports and other physical activity without restriction, due to the guidelines and safety measures for competitive sports in Austria [19]. In SF classes, students completed a two-hour sports lesson three times weekly, mostly indoors.

Students attending the GB were allowed to carry out physical activities and sports only under strict restrictions in compliance with existing COVID-19 mitigation measures [19].

At the onset of data collection, 616 students (421 in GB and 195 in SF) were attending secondary school education at the campus. They and their parents were asked to provide information about SARS-CoV-2 infection during the period of interest. A total of 213 legal guardians (for children < 14 years) and 401 students (≥14 years) gave written consent to participate in the study. Two students (0.3%) did not want to participate in the study, both of them from the GB classes.

### 2.3. Definition of Different Periods Based on Dominant Virus Variants and Mitigation Measures

In Austria, a detailed school safety concept was mandated by the Federal Ministry of Education, Science, and Research in August 2021 for the 2021/22 school year in order to ensure largely unrestricted school operations during the COVID-19 pandemic. Three safety levels with different mitigation measures were defined to be adapted to the regional SARS-CoV-2 infection situation [19].

In September and October 2021, very low seven-day incidence of SARS-CoV-2 infections occurred in the region around the school campus. Therefore, after a three-week security phase (with one polymerase chain reaction (PCR) test and two rapid antigen tests weekly), security level 1 was enacted (students had the option of voluntary testing at school via a rapid antigen test).

From 2 November 2021 to 28 February 2022, security level 3 was enacted, meaning that students were required to repeatedly perform a SARS-CoV-2 screening test three times a week (two polymerase chain reaction (PCR) tests and one rapid antigen test), to keep a safety distance of 1 m, and to wear an FFP-2 mask throughout the school building. If infection with SARS-CoV-2 was detected by PCR testing, students had to stay at home for 10 days. Subsequently, students who tested positive were excluded from further testing for the following 90 days [20,21]. From 28 February 2022 onwards, the strict mitigation measures were relaxed and students were no longer requested to wear FFP-2 masks in classrooms. However, PCR and antigen testing were continued (two polymerase chain reaction (PCR) tests and one rapid antigen test) [22].

During the study period, two variants of SARS-CoV-2 circulated in Austria that were classified as variants of concern by the World Health Organization (WHO) [23]. The Delta variant B.1.617.2 was predominant until December 2021 [24,25], while from January 2022 onwards the Omicron variant BA.1 [26] was the dominant variant [24,25].

Based on the different mitigation measures imposed and the dominant variants in the various periods, the study period was divided into four periods of relatively equal length (see also Appendix A):

P1 (13 September 2021 to 31 October 2021): the predominant SARS-CoV-2 variant = Delta (B.1.617.2); the low seven-day incidence of SARS-CoV-2 infections, security phase, and security level 1;

P2 (1 November 2021 to 31 December 2021): the predominant SARS-CoV-2 variant = Delta (B.1.617.2); a high seven-day incidence of SARS-CoV-2 infections, security level 3;

P3 (1 January 2022 to 28 February 2022): the predominant SARS-CoV-2 variant = Omicron (BA.1); a very high seven-day incidence of SARS-CoV-2 infections, security level 3;

P4 (1 March 2022 to 30 April 2022): the predominant SARS-CoV-2 variant = Omicron (BA.2/BA.3); a very high seven-day incidence of SARS-CoV-2 infections, and relaxed security level 3 without wearing FFP2 masks in classes.

### 2.4. Procedures

In May and June 2022, the students provided information on whether and when they had had an infection with SARS-CoV-2 detected by PCR test between 13 September 2021 and 30 April 2022.

Dichotomous data (the detection of SARS-CoV-2 by PCR testing: yes or no) were generated for each day of the study period for all participants. In a second step, potential secondary cases (infection) in the class were identified to estimate the secondary attack rate considering the current knowledge about mean generation time (GT).

Mean GT is the time interval between a reported infection of a primary case (infector) and potential secondary cases (infected) [27]. Previous studies have reported different GTs for different variants of SARS-CoV-2 [28,29,30,31]. Zhang et al. [28] reported a mean GT of 2.9 days for B.1.617.2, while Hart et al. [29] reported a mean intrinsic GT of 4.7 days and a mean household GT of 3.2 days for this variant. Ito et al. [30] reported a reduction in mean GT for the Omicron variant BA.1 compared with the Delta variant B.1.617.2 (GT for BA.1 = 0.44 to 0.46 times Delta), and similar results were reported by Manica et al. [31].

Based on these reports and the fact that GTs for potential secondary cases (infections) are difficult to measure [27], different intervals (2, 4, 6, or 8 days) for mean GT were used in our analyses. For all reported SARS-CoV-2 infections, potential secondary cases (infections) in the same classroom were identified for each assumed GT (GT = 2, 4, 6, or 8 days) and dichotomous data (yes or no) were generated for each day and student.

### 2.5. Outcomes

The primary outcomes were SARS-CoV-2 infection dynamics (a cumulative percentage of students with a SARS-CoV-2 infection), which were compared between the SF and GB school classes. Using reported SARS-CoV-2 infections, daily seven-day class incidence was calculated by dividing the number of positive test results in the previous seven days by the number of students attending class and extrapolating the obtained seven-day class density to 100,000 individuals.

As mentioned, potential secondary cases in school classes were identified using different GTs (2, 4, 6, or 8 days), and odds ratios were calculated for students in SF and GB classes.

Secondary analyses were performed for subgroups based on school grade (middle school (M.S.): age 10–16 years (mean 13.1 [95%CI = 12.9–13.2 years]); four-year high school (H.S.): age 14–20 years (mean 16.9 [95%CI = 16.8–17.1 years])) and sex (boys; girls).

### 2.6. Statistical Analysis

Descriptive statistics were calculated; continuous variables are expressed as mean (M) and standard deviation (SD), and categorial variables as absolute values (n) and percentages (%), and no data imputation was performed.

We used a chi-square test (X^2^) or a Fisher’s exact test where appropriate to test for differences in cumulative percentages of students with a SARS-CoV-2 infection in SF and GB classes.

Phi coefficients (φ) were calculated to estimate the correlation strength between stringent mitigation measures in the SF and GB school classes.

Additionally, a binary logistic regression analysis was performed to assess the relationship between class membership (SF or GB) and the cumulative percentage of students with a SARS-CoV-2 infection (CPI) or the potential SARS-CoV-2 cases (infections) in classroom settings. Coefficients obtained from binary logistic regression were expressed as odds ratios (OR) with 95% confidence intervals. Due to a lack of variance homogeneity, we used the Welch test to analyze differences in seven-day incidences.

All tests were two-sided, with a *p*-value of <0.05 considered statistically significant. Phi (φ) according to Cohen [32] was used to determine the effect size (≥0.1, small; ≥0.3, medium; and ≥0.5, large). All statistical calculations were performed using SPSS Version 27 (IBM Corp. Released 2020. IBM SPSS Statistics for Windows, IBM Corp., Armonk, NY, USA).

## 3. Results

Among the 614 students (age: 15.1 ± 2.3 years, 43.2% female) included in this analysis, 195 students (31.8%) attended school classes with a sports focus (SF) and were allowed to participate in sports without restrictions during the study period. Significantly fewer girls were in SF classes than in GB classes (SF: ♀ = 25.1%, GB: ♀ = 51.6%; *p* ≤ 0.001). Students in SF classes were slightly younger than those in GB classes (SF: 14.8 years; GB: 15.3 years; *p* = 0.006) (Appendix A).

In total, 76 students (12.4%) had been infected with SARS-CoV-2 before the study period (before 13 September 2021). More previous infections were reported by older students (H.S.) in SF classes than by students in parallel GB classes (H.S. = SF: 22.9%; GB: 11.8%; and *p* = 0.011). No differences in previous infections were reported by younger students (M.S. = SF: 10.1%; GB: 8.9%; *p* = 0.75) (Appendix A).

### 3.1. Infection Status—Changes over Time

#### 3.1.1. CPI—Cumulative Percentage of Students with a SARS-CoV-2 Infection

No significant differences were found in the CPI by the end of the time period 1 (P1) between SF and GB classes, and overall the CPI levels were very low (SF: 2.6%; GB 1.2%; *p* = 0.30; and φ = 0.05) (Table 1, Figure 1).

In time period 2 (P2), the CPI increased and moderate differences were observed (end of December 2021: SF: 16.4%; GB: 9.1%; *p* = 0.008, φ = 0.11). For students in SF school classes, the odds of becoming infected with SARS-CoV-2 increased during P2 (the end of November 2021: OR = 2.64 (95% CI, 1.51–4.62); the end of December 2021: OR = 1.97 (95% CI, 1.19–3.26)). ORs were higher for younger (M.S.) students (the end of November 2021: OR = 3.59 (95% CI, 1.51–8.54); the end of December 2021: OR = 2.59 (95% CI, 1.24–5.40)) than older (H.S.) students (the end of November 2021: OR = 2.13 (95% CI, 1.00–4.52); the end of December 2021: OR = 1.53 (95% CI, 0.75–3.12)) (Table 1, Figure 1).

CPI increased dramatically in time period 3 (P3) due to the Omicron wave. By the end of February 2022, greatly increased CPI in both branches went along with a significant difference between the school focuses (SF: 60.6%; GB: 36.8%; *p* < 0.001; and φ = 0.23). At the end of February 2022, students in the SF school classes exhibited a 2.6-fold increased probability of having been infected with SARS-CoV-2 (end of February 2022: OR = 2.61 (95% CI, 1.84–3.69)) (Table 1, Figure 1).

By the end of March 2022, this difference decreased, with students in GB school classes showing significantly higher CPI compared to the end of the previous month (SF: 61.5%; GB: 53.2%; *p* = 0.018, φ = 0.10). This trend continued until the end of April 2022 (SF: 63.1%; GB: 54.7%; *p* = 0.037, φ = 0.09) and could also be observed in different school grades (M.S. and H.S.) (Table 1, Figure 1).

Detailed information on different CPI trends in subgroups are reported in the Appendix A, Appendix A.

#### 3.1.2. Mean 7-Day Incidence for Infection with SARS-CoV-2 (M7D-I SARS-CoV-2)

M7D-I SARS-CoV-2 showed significant differences (*p* < 0.001) between school focuses in all four time periods. In P1, P2, and P3, higher (*p* < 0.001) M7D-I SARS-CoV-2 was observed in SF school classes. During P4, the situation reversed, and significantly higher (*p* < 0.001) M7D-I SARS-CoV-2 was detected in GB school classes (Table 2, Figure 2).

Detailed information on the trends in M7D-I SARS-CoV-2 is reported in the Appendix A.

#### 3.1.3. Potential SARS-CoV-2 Infection in School Classes (PI-SARS-CoV-2)

When PI-SARS-CoV-2 was analyzed, small effects were found in P2, P3, and P4 regardless of which hypothesized GT (mean generation time) had been used. Due to the limited number of participants, GT for two days during P1, P2, and P4 partially identified insufficient numbers of potential infection cases. Regardless of which GT was used, there were no significant changes in the overall conclusion of which school branch was more likely to be infected with SARS-CoV-2 (Table 3).

In P2 and P3, the odds of PI-SARS-CoV-2 were higher in SF classes (hypothesized GT for four days = P2: OR = 2.78 (95% CI, 1.08–7.15); P3: OR = 4.03 (95% CI, 2.56–6.31)). These changes in P4, where the odds of PI-SARS-CoV-2 were significantly reduced in SF classes, were in contrast to those in P2 and P3 (hypothesized GT for four days = P4: OR = 0.05 (95% CI, 0.01–0.39)) (Table 3).

Additional results for differentially hypothesized GTs are reported in the Appendix A.

## 4. Discussion

To our knowledge, our study is the first to assess the impact of wearing face masks in classroom settings. Our results show that under otherwise strict mitigation measures, students allowed to participate in sports without restriction (SF group) were at a significantly higher risk of being infected by SARS-CoV-2 than those following general strict mitigation measures at school. This is in line with the results of studies reporting the clear efficacy of wearing FFP2 or any other face masks, with both leading to a reduction in viral (variant) spreading [33,34].

To the best of our knowledge, the long-term effects of continued mask wearing have been investigated in only a few studies to date [35,36,37,38]. These suggest that the continued wearing of face masks may also have negative consequences, and these must be carefully balanced against the advantage of reduced infection rates. In this context, the question arises of whether the consequent routine use of face masks is able to reduce long-term infection rates or merely postpones numerous infections. The latter could reduce the usefulness of long-term mask obligations, together with the reduced pathogenicity of the virus variant that is circulating.

In late February 2022, an increasing number of studies reported that Omicron variants led to less severe disease, and reduced hospitalization rates. In response, stringent mitigation measures were relaxed worldwide [39,40]. Our data show that this (at least in school settings) apparently led to “catch-up” infections in those thus far protected from infection with SARS-CoV-2. By the end of our data collection (April 2022), only a small difference in cumulative infection rates was observed between students with strict mask wearing rules during the high-incidence period and those without.

This leads to the question when, for whom, and which face masks should be made obligatory in school settings, and especially for how long. These questions must be answered by medical research and should no longer be a matter of individual/local/regional decision-making. Although no major physical/physiological negative side effects of face mask use have been proven for healthy infants so far [36], continued or even long-term use may have a negative impact on long-term psychosocial health in growing and developing individuals [13,14,15,41,42]. Similarly, future research should investigate whether and, if so, what negative physical effects can be triggered by long-term wearing of a face mask [43]. In an attempt to avoid viral transmission whenever and wherever possible, this aspect was seemingly underrepresented in systematic COVID-19 research. Our study demonstrates that wearing face masks is effective at reducing viral transmission. This benefit is, however, limited in the long run. Thus, the benefit of viral transmission reduction must be carefully balanced against potential negative effects on mental and social health.

Due to the limited number of school classes without strict mitigation measures during the school year 2021/2022, no random representative sample selection was possible. However, the sample size was large and the participation rate was high (99.7%).

Another limitation of our study is that the actual chain of SARS-CoV-2 transmissions cannot be detected flawlessly in practice. In fact, transmission could also have taken place outside school by personal contacts during leisure time. This limitation, however, holds true for almost all studies dealing with secondary attack estimations.

Regardless of the limitations, our results clearly show the “catch-up” of infections in response to the relaxation of strict mitigation measures. Thus, the temporary obligation of face mask use in schools can postpone a number of secondary infections but has a very limited effect on the long-term avoidance of these infections. This finding should be considered regarding further decisions concerning obligatory face mask use in school settings.

## 5. Conclusions

Our study shows that a number of infections with SARS-CoV-2 are delayed, but they cannot be prevented in the long run by wearing face masks. Therefore, the obligatory use of face masks in schools may be understood as an epidemiological measure to flatten SARS-CoV-2 peaks rather than to protect individuals. Since healthy school children are rarely severely affected by COVID, on the one hand, but may experience negative psychosocial consequences, on the other hand, by continued face mask use, the advantage of (temporarily) reduced virus transmission must be carefully balanced against the potential negative consequences on psychosocial development and mental health.

## Figures and Tables

**Figure 1 ijerph-19-13511-f001:**
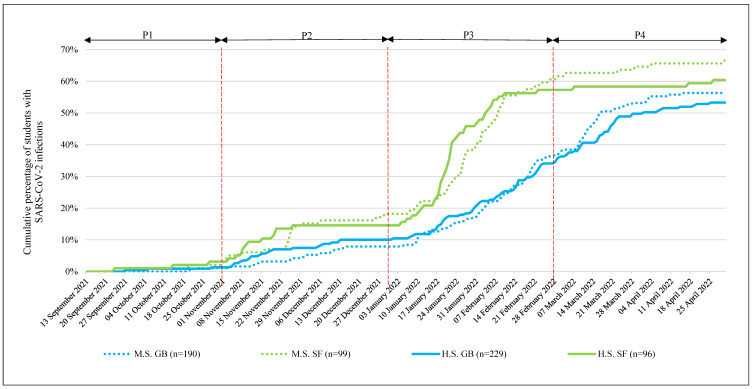
Cumulative percentage of students with SARS-CoV-2 infections. SARS-CoV-2 = severe acute respiratory syndrome coronavirus 2, P1 = period 1 (13 September 2021 to 31 October 2021), P2 = period 2 (1 November 2021 to 31 December 2021), P3 = period 3 (1 January 2022 to 28 February 2022), P4 = period 4 (1 March 2022 to 30 April 2022), M.S. = middle school (children aged 13.1 ± 1.3 years old), H.S. = 4-year high school (children aged 16.9 ± 1.2 years old), GB = students in school classes in a general school branch, SF = students in school classes with sport focus.

**Figure 2 ijerph-19-13511-f002:**
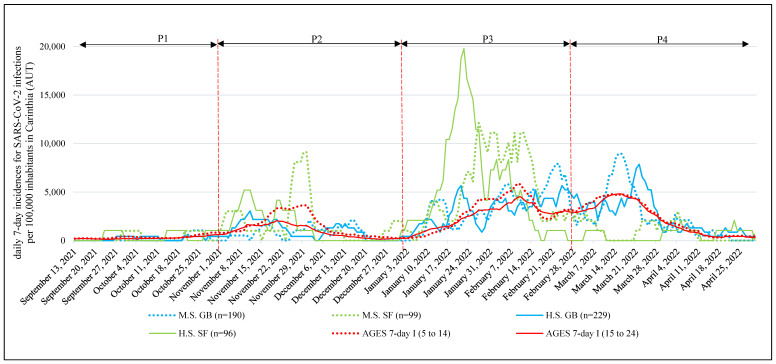
Differences in 7-day incidence of SARS-CoV-2 infections per 100,000 inhabitants in Carinthia (AUT) and among students of secondary school age with or without a sport focus. SARS-CoV-2 = severe acute respiratory syndrome coronavirus 2, P1 = period 1 (13 September 2021 to 31 October 2021), P2 = period 2 (1 November 2021 to December 2021), P3 = period 3 (1 January 2022 to 28 February 2022), P4 = period 4 (1 March 2022 to 30 April 2022), M.S. = middle school (children aged 13.1 ± 1.3 years old), H.S. = 4-year high school (children aged 16.9 ± 1.2 years old), GB = students in school classes in a general school branch, SF = students in school classes with sport focus, AGES = Austrian Agency for Health and Food Safety (https://covid19-dashboard.ages.at/, accessed on 5 September 2022), 7-day I (5 to 14) = 7 days incidence for SARS-CoV-2 infections per 100,000 inhabitants in Carinthia (AUT) for students aged 5 to 14 years, 7-day I (15 to 24) = 7 days incidence for SARS-CoV-2 infections per 100,000 inhabitants in Carinthia (AUT) for people aged 15 to 24 years, AUT = Austria.

**Table 1 ijerph-19-13511-t001:** Cumulative percentage of students with a SARS-CoV-2 infection: students in classes in GB vs. students in SF classes.

Variable	Time Period	All (*n* = 614)		*p* Value	φ	OR (95% CI)
		GB (*n* = 419)	SF (*n* = 185)			
CU% SARS-CoV-2, …, No. [%]	…September 2021 (P1)	1 (0.2%)	2 (1.0%)	0.24	0.05	4.33 (0.39 to 48.06)
	…October 2021 (P1)	5 (1.2%)	5 (2.6%)	0.30	0.05	2.17 (0.62 to 7.62)
	…November 2021 (P2)	26 (6.2%)	29 (14.9%)	<0.001	0.14	2.64 (1.51 to 4.62)
	…December 2021 (P2)	38 (9.1%)	32 (16.4%)	0.008	0.11	1.97 (1.19 to 3.26)
	…January 2022 (P3)	81 (19.3%)	85 (43.6%)	<0.001	0.25	3.22 (2.22 to 4.68)
	…February 2022 (P3)	149 (35.6%)	115 (59.0%)	<0.001	0.22	2.61 (1.84 to 3.69)
	…March 2022 (P4)	215 (51.3%)	120 (61.5%)	0.018	0.10	1.52 (1.07 to 2.15)
	…April 2022 (P4)	229 (54.7%)	124 (63.6%)	0.037	0.09	1.45 (1.02 to 2.06)
		**M.S. (*n* = 289)**				
		**GB** **(*n* = 190)**	**SF** **(*n* = 99)**			
CU% SARS-CoV-2, …, No. [%]	…September 2021 (P1)	0 (0.0%)	1 (1.0%)	0.34	0.08	INC
	…October 2021 (P1)	2 (1.1%)	2 (2.0%)	0.61	0.04	1.94 (0.27 to 13.97)
	…November 2021 (P2)	9 (4.7%)	15 (15.2%)	0.002	0.18	3.59 (1.51 to 8.54)
	…December 2021 (P2)	15 (7.9%)	18 (18.2%)	0.009	0.15	2.59 (1.24 to 5.40)
	…January 2022 (P3)	33 (17.4%)	40 (40.4%)	<0.001	0.25	3.23 (1.86 to 5.59)
	…February 2022 (P3)	70 (36.8%)	60 (60.6%)	<0.001	0.23	2.64 (1.60 to 4.35)
	…March 2022 (P4)	101 (53.2%)	64 (64.6%)	0.06	0.11	1.61 (0.98 to 2.66)
	…April 2022 (P4)	107 (56.3%)	66 (66.7%)	0.09	0.10	1.55 (0.94 to 2.58)
		**H.S. (*n* = 325)**				
		**GB** **(*n* = 229)**	**SF** **(*n* = 96)**			
CU% SARS-CoV-2, …, No. [%]	…September 2021 (P1)	1 (0.4%)	1 (1.0%)	0.50	0.04	2.40 (0.15 to 38.77)
	…October 2021 (P1)	3 (1.3%)	3 (3.1%)	0.37	0.06	2.43 (0.48 to 12.26)
	…November 2021 (P2)	17 (7.4%)	14 (14.6%)	0.045	0.11	2.13 (1.00 to 4.52)
	…December 2021 (P2)	23 (10.0%)	14 (14.6%)	0.24	0.07	1.53 (0.75 to 3.12)
	…January 2022 (P3)	48 (21.0%)	45 (46.9%)	<0.001	0.26	3.32 (1.99 to 5.55)
	…February 2022 (P3)	79 (34.5%)	55 (57.3%)	<0.001	0.21	2.55 (1.56 to 4.15)
	…March 2022 (P4)	114 (49.8%)	56 (58.3%)	0.16	0.08	1.41 (0.87 to 2.29)
	…April 2022 (P4)	122 (53.3%)	58 (60.4%)	0.24	0.07	1.34 (0.83 to 2.17)

Data are No. (%), M.S. = middle school (students aged 13.1 ± 1.3 years old), H.S. = 4-year high school (students aged 16.9 ± 1.2 years old), GB = students in school classes in a general school branch, SF = students in school classes with sport focus, φ = effect size Phi, OR = odds ratio (reference group: students without sport focus), CI = confidence interval, CU% SARS-CoV-2 = cumulative percentage of students with SARS-CoV-2 infections at the end of…, SARS-CoV-2 = severe acute respiratory syndrome coronavirus 2, P1 = period 1 (13 September 2021 to 31 October 2021) P2 = period 2 (1 November 2021 to 31 December 2021), P3 = period 3 (1 January 2022 to 28 February 2022), P4 = period 4 (1 March 2022 to 30 April 2022), INC = insufficient number of potential infection cases.

**Table 2 ijerph-19-13511-t002:** Mean 7-day incidences in classrooms for SARS-CoV-2 infections per 100,000 inhabitants: students in GB classes vs. students in classes with SF.

Variable	Time Period	Mean AGES 7-day I (5 to 14)	M.S. (*n* = 289)		t ^a^	*p* Value	*p*-lvl	M.D.	95% CI	
			GB (*n* = 190)	SF (*n* = 99)						
									Lo	Up
Mean 7-day I (CR)	P1	370.02	118.85 ± 202.19	251.62 ± 251.08	−4.549	<0.001	***	−132.77	−190.40	−75.15
	P2	1521.82	783.81 ± 732.41	1903.99 ± 564.43	−14.412	<0.001	***	−1120.17	−1273.26	−967.08
	P3	2797.19	3593.86 ± 1502.88	5319.90 ± 653.95	−13.558	<0.001	***	−1726.04	−1976.64	−1475.44
	P4	2361.77	2372.95 ± 1155.04	906.38 ± 870.70	12.105	<0.001	***	1466.57	1227.96	1705.19
		**Mean AGES 7-day I (15 to 24)**	**H.S. (*n* = 325)**							
			**GB (*n* = 229)**	**SF (*n* = 96)**						
Mean 7-day I (CR)	P1	268.88	196.43 ± 335.04	404.16 ± 507.97	−3.685	<0.001	***	−207.73	−319.25	−96.21
	P2	962.70	1030.43 ± 768.68	1370.41 ± 1145.24	−2.668	<0.001	***	−339.98	−592.08	−87.88
	P3	2497.02	3141.79 ± 994.91	5324.14 ± 1282.66	−14.897	<0.001	***	−2182.35	−2471.88	−1892.82
	P4	2185.20	2488.14 ± 1203.69	602.6 ± 582.11	18.992	<0.001	***	1885.55	1690.21	2080.88

^a^ = Due to lack of variance homogeneity, we used the Welch test. Data are mean (SD), SD = standard deviation; AGES = Austrian Agency for Health and Food Safety, 7-day I (5 to 14) = 7 days incidence for SARS-CoV-2 infections per 100,000 inhabitants in Carinthia (AUT) for children aged 5 to 14 years (https://covid19-dashboard.ages.at/, accessed on 5 September 2022), 7-day I (15 to 24) = 7 days incidence for SARS-CoV-2 infections per 100,000 inhabitants in Carinthia (AUT) for adolescents aged 15 to 24 years (https://covid19-dashboard.ages.at/, accessed on 5 September 2022), AUT = Austria, M.S. = middle school (students aged 13.1 ± 1.3 years old), H.S. = 4-year high school (students aged 16.9 ± 1.2 years old), GB = students in school classes in a general school branch, SF = students in school classes with sport focus, t = test statistic Welch test; *p*-lvl (*p* value level) *** = *p* < 0.001, M.D. = mean difference, CI = confidence interval of the difference, Lo = lower, Up = upper, I (CR) = incidence in classroom, CR = classroom, P1 = period 1 (13 September to 31 October 2021), P2 = period 2 (1 November to 31 December 2021), P3 = period 3 (1 January to 28 February 2022), P4 = period 4 (1 March and 30 April 2022).

**Table 3 ijerph-19-13511-t003:** Potential SARS-CoV-2 infection in school classes SF vs. GB, using different hypothesized generation times.

Time Period	School Grade	Category	HTPPI GT 2D	HTPPI GT 4D	HTPPI GT 6D	HTPPI GT 8D
			OR (95%CI)	OR (95%CI)	OR (95%CI)	OR (95%CI)
P1	All (*n* = 614)	GB (*n* = 419)	INC	INC	INC	INC
		SF (*n* = 185)				
	M.S. (*n* = 289)	GB (*n* = 190)	INC	INC	INC	INC
		SF (*n* = 99)				
	H.S. (*n* = 325)	GB (*n* = 229)	INC	INC	INC	INC
		SF (*n* = 96)				
P2	All (*n* = 614)	GB (*n* = 419)	5.16 (1.32 to 20.19)	2.78 (1.08 to 7.15)	2.92 (1.26 to 6.79)	2.87 (1.28 to 6.44)
		SF (*n* = 185)				
	M.S. (*n* = 289)	GB (*n* = 190)	INC	12.19 (1.45 to 02.76)	4.02 (0.98 to 16.44)	4.74 (1.20 to 8.76)
		SF (*n* = 99)				
	H.S. (*n* = 325)	GB (*n* = 229)	1.60 (0.26 to 9.75)	1.38 (0.39 to 4.82)	2.49 (0.85 to 7.32)	2.17 (0.77 to 6.17)
		SF (*n* = 96)				
P3	All (*n* = 614)	GB (*n* = 419)	4.02 (2.38 to 6.81)	4.03 (2.56 to 6.31)	3.78 (2.45 to 5.83)	3.30 (2.18 to 4.99)
		SF (*n* = 185)				
	M.S. (*n* = 289)	GB (*n* = 190)	4.24 (2.00 to 8.99)	3.77 (1.96 to 7.24)	3.50 (1.88 to 6.53)	3.01 (1.64 to 5.51)
		SF (*n* = 99)				
	H.S. (*n* = 325)	GB (*n* = 229)	3.79 (1.81 to 7.93)	4.29 (2.29 to 8.01)	4.08 (2.23 to 7.43)	3.64 (2.06 to 6.42)
		SF (*n* = 96)				
P4	All (*n* = 614)	GB (*n* = 419)	INC	0.05 (0.01 to 0.39)	0.11 (0.04 to 0.37)	0.14 (0.05 to 0.38)
		SF (*n* = 185)				
	M.S. (*n* = 289)	GB (*n* = 190)	INC	0.10 (0.01 to 0.79)	0.15 (0.04 to 0.65)	0.21 (0.06 to 0.70)
		SF (*n* = 99)				
	H.S. (*n* = 325)	GB (*n* = 229)	INC	0.10 (0.01 to 0.79)	0.08 (0.01 to 0.56)	0.07 (0.01 to 0.50)
		SF (*n* = 96)				

SARS-CoV-2 = severe acute respiratory syndrome coronavirus 2, HTPPI = hypothetical time period assumed for potential infections, GT = generation time, D = days, OR = odds ratio for potential SARS-CoV-2 infection in school classes SF vs. GB, calculated with a binary logistic regression by using different hypothesized generation times (with 1 for students in GB classes), CI = confidence interval, P1 = period 1 (13 September 2021 to 31 October 2021), P2 = period 2 (1 November 2021 to 31 December 2021), P3 = period 3 (1 January 2022 to 28 February 2022), P4 = period 4 (1 March 2021 to 30 April 2022), M.S. = middle school (children aged 13.1 ± 1.3 years old), H.S. = 4-year high school (children aged 16.9 ± 1.2 years old), GB = children in school classes in a general branch, SF = children in school classes with sport focus, *n* = study population, INC = insufficient number of potential infection cases.

## Data Availability

The data presented in this study are available on request from the corresponding author. The data are not publicly available due to privacy/ethical restrictions.

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
