# Peer review of "Effects of Wearing FFP2 Masks on SARS-CoV-2 Infection Rates in Classrooms"

_ijerph, 2022, doi:10.3390/ijerph192013511_

Round 1

Reviewer 1 Report

The manuscript submitted for review is interesting, however, I have some considerations:

The summary seems somewhat confusing. In the attempt to give a large amount of information, the reading becomes a bit dense. I suggest improving the wording in this part.

Remember that many times, the abstract is the only thing that colleagues read.

The introduction is quite succinct and directed towards the research question, which is good; However, information is needed on two important points: the impact that exercise has on the transmission of the disease and the need to assess all the edges that the use of masks in covid may have. I suggest using this reference: Well-Designed Studies are Needed to Assess Adverse Effects on Healthy Lung Function after Long-Term Face Masks Usage. https://doi.org/10.54034/mic.e1222.

The methods are fairly well described. In line 62 the word "branch" should be removed after the acronym.

It must adequately support what is the reason for dividing the study time into these four periods (in my opinion the time used is quite narrow), since there are in the subsequent statistics, there are results where no analysis is offered due to the insufficient number of cases.

The discussion requires further evaluation of all of its results, since it does not adequately discuss the interrelationship of mask use with other potentially confounding factors. I suggest considering this study: Analysis of some factors and COVID-19 mortality in the population of 0 to 24 years in 29 countries: open schools could be a protection. https://doi.org/10.54034/mic.e1480.

The answer they give about the methodological considerations is crucial in the importance of the manuscript.

Author Response

Answer for Reviewer 1

Thank you very much for the supportive assessment of our paper.  Please find below our responses to your comments.

The manuscript submitted for review is interesting, however, I have some considerations:

C1: The summary seems somewhat confusing. In the attempt to give a large amount of information, the reading becomes a bit dense. I suggest improving the wording in this part.

Remember that many times, the abstract is the only thing that colleagues read.

A1: Thanks for the feedback, we have made minor changes in the abstract.

C2: The introduction is quite succinct and directed towards the research question, which is good; However, information is needed on two important points: the impact that exercise has on the transmission of the disease and the need to assess all the edges that the use of masks in covid may have. I suggest using this reference: Well-Designed Studies are Needed to Assess Adverse Effects on Healthy Lung Function after Long-Term Face Masks Usage. https://doi.org/10.54034/mic.e1222.

A2: Thank you for the reference. We have included the study in our report. However, not in the introduction section but in the discussion ( see line 338).

We would like to note that also on the part of the authors it has already been discussed what could be potentially harmful effects related to long-term mask wearing and we have also already worked out detailed concepts on which methods could be used to implement such studies. In this context, our opinions are in line with your suggestions of the recommended study and we think that this kind of research is urgently needed for future health-related political decisions.

C3: The methods are fairly well described. In line 62 the word "branch" should be removed after the acronym.

A3: Thank you very much for bringing this error to our attention, we have corrected it.

C4: It must adequately support what is the reason for dividing the study time into these four periods (in my opinion the time used is quite narrow), since there are in the subsequent statistics, there are results where no analysis is offered due to the insufficient number of cases.

A4: From the authors' point of view, the reasons for the division into the 4 periods have been sufficiently reported. In particular, the supplements in Table S1 report in detail on the different prevailing conditions that made the division into these periods necessary. The key factors here were the significantly different restrictions imposed in the individual periods, the high differences in case numbers (SARS-CoV-2 infection numbers), and the associated different dominant subvariants of the SARS-CoV-2 virus. Therefore, for the team of authors, the classification in the present periods is the only possible one, although a different classification would of course have been helpful for the analysis due to the low case numbers.

C5: The discussion requires further evaluation of all of its results, since it does not adequately discuss the interrelationship of mask use with other potentially confounding factors. I suggest considering this study: Analysis of some factors and COVID-19 mortality in the population of 0 to 24 years in 29 countries: open schools could be a protection. https://doi.org/10.54034/mic.e1480. The answer they give about the methodological considerations is crucial in the importance of the manuscript.

A5: Thank you for bringing this study to our attention, which reports very interesting results. In particular, this study is about the positive benefits of school openings during the COVID-19 pandemic. However, since in our report, possible school closures do not play a role, the authors do not want to include this study in this report. However, we will keep the study in evidence for future reports and thank you for pointing it out.

Reviewer 2 Report

Dear authors,

Congratulations on your work. Very interesting and innovative. 

I would suggest to read the article Schlegtendal, A.; Eitner, L.; Falkenstein, M.; Hoffmann, A.; Lücke, T.; Sinningen, K.; Brinkmann, F. To Mask or Not to Mask—Evaluation of Cognitive Performance in Children Wearing Face Masks during School essons (MasKids). Children 2022, 9, 95. https://doi.org/10.3390/children9010095 and to use it in the Discussion session.

Author Response

Answer for Reviewer 2

Thank you very much for the supportive assessment of our paper.  Please find below our responses to your comments.

C1: Congratulations on your work. Very interesting and innovative.

A1: Thank you for these very motivating words

C2: I would suggest to read the article Schlegtendal, A.; Eitner, L.; Falkenstein, M.; Hoffmann, A.; Lücke, T.; Sinningen, K.; Brinkmann, F. To Mask or Not to Mask—Evaluation of Cognitive Performance in Children Wearing Face Masks during School essons (MasKids). Children 2022, 9, 95. https://doi.org/10.3390/children9010095 and to use it in the Discussion session.

A2: Thank you for the reference. We have included the study in our report. However, not in the introduction section but in the discussion ( see line 336).

Round 2

Reviewer 1 Report

The way in which the authors have addressed my comments is enough for me, although I do not agree with the time they have used, the answer they have given is satisfactory and valid.